# Elucidating Hexanucleotide Repeat Number and Methylation within the X-Linked Dystonia-Parkinsonism (XDP)-Related SVA Retrotransposon in *TAF1* with Nanopore Sequencing

**DOI:** 10.3390/genes13010126

**Published:** 2022-01-11

**Authors:** Theresa Lüth, Joshua Laβ, Susen Schaake, Inken Wohlers, Jelena Pozojevic, Roland Dominic G. Jamora, Raymond L. Rosales, Norbert Brüggemann, Gerard Saranza, Cid Czarina E. Diesta, Kathleen Schlüter, Ronnie Tse, Charles Jourdan Reyes, Max Brand, Hauke Busch, Christine Klein, Ana Westenberger, Joanne Trinh

**Affiliations:** 1Institute of Neurogenetics, University of Luebeck, 23538 Luebeck, Germany; theresa.lueth@neuro.uni-luebeck.de (T.L.); joshua.lass@student.uni-luebeck.de (J.L.); susen.schaake@neuro.uni-luebeck.de (S.S.); jelena.pozojevic@neuro.uni-luebeck.de (J.P.); norbert.brueggemann@neuro.uni-luebeck.de (N.B.); kathleen.schlueter@student.uni-luebeck.de (K.S.); ronnie.tse@neuro.uni-luebeck.de (R.T.); charles.reyes@neuro.uni-luebeck.de (C.J.R.); max.brand@student.uni-luebeck.de (M.B.); christine.klein@neuro.uni-luebeck.de (C.K.); ana.westenberger@neuro.uni-luebeck.de (A.W.); 2Medical Systems Biology Division, Luebeck Institute of Experimental Dermatology, University of Luebeck, 23538 Luebeck, Germany; Inken.Wohlers@uni-luebeck.de (I.W.); Hauke.Busch@uni-luebeck.de (H.B.); 3Institute for Cardiogenetics, University of Luebeck, 23538 Luebeck, Germany; 4Department of Neurosciences, College of Medicine, Philippine General Hospital, University of the Philippines Manila, Manila 1000, Philippines; rgjamora@up.edu.ph; 5Department of Neurology and Psychiatry, The Hospital Neuroscience Institute, University of Santo Tomas, Manila 1008, Philippines; rlrosales@ust.edu.ph; 6Department of Neurology, University of Luebeck, 23538 Luebeck, Germany; 7Section of Neurology, Department of Internal Medicine, Chong Hua Hospital, Cebu City 6000, Philippines; gerardsaranza@gmail.com; 8Department of Neurosciences, Movement Disorders Clinic, Makati Medical Center, Makati 1229, Philippines; ciddiesta@gmail.com

**Keywords:** X-linked dystonia-parkinsonism, nanopore sequencing, repeat motif, CpG methylation

## Abstract

Background: X-linked dystonia-parkinsonism (XDP) is an adult-onset neurodegenerative disorder characterized by progressive dystonia and parkinsonism. It is caused by a SINE-VNTR-Alu (SVA) retrotransposon insertion in the *TAF1* gene with a polymorphic (*CCCTCT*)_n_ domain that acts as a genetic modifier of disease onset and expressivity. Methods: Herein, we used Nanopore sequencing to investigate SVA genetic variability and methylation. We used blood-derived DNA from 96 XDP patients for amplicon-based deep Nanopore sequencing and validated it with fragment analysis which was performed using fluorescence-based PCR. To detect methylation from blood- and brain-derived DNA, we used a Cas9-targeted approach. Results: High concordance was observed for hexanucleotide repeat numbers detected with Nanopore sequencing and fragment analysis. Within the SVA locus, there was no difference in genetic variability other than variations of the repeat motif between patients. We detected high CpG methylation frequency (MF) of the SVA and flanking regions (mean MF = 0.94, SD = ±0.12). Our preliminary results suggest only subtle differences between the XDP patient and the control in predicted enhancer sites directly flanking the SVA locus. Conclusions: Nanopore sequencing can reliably detect SVA hexanucleotide repeat numbers, methylation and, lastly, variation in the repeat motif.

## 1. Introduction

β X-linked dystonia-parkinsonism (XDP) is a neurodegenerative movement disorder and its phenomenology was first described in the literature in 1976 [1]. Patients originate mainly from the Philippines or are of Filipino descent and mainly aggregate on the island of Panay. A known family history of the disease is present for ~94% of the patients. XDP originated through a founder mutation approximately 1000 years ago [2]. The disease is characterized by dystonic movements and postures as well as parkinsonism due to an insertion of the retrotransposon SINE-VNTR-Alu (SVA) in intron 32 of the *TAF1* (*TATA-binding protein-associated factor 1*) gene [3,4].

The *TAF1* SVA insertion has five domains. At the 5′ end, there is a hexanucleotide repeat domain, which consists of the repeat sequence (*CCCTCT*)_n_ [5]. This hexanucleotide repeat (*CCCTCT*)_n_ domain varies in repeat numbers among patients, ranging from 30 to 55. The repeat number is inversely correlated with age at onset and disease severity [5,6]. In addition, somatic mosaicism has been observed, with a higher median number of repeats detected in the cerebellum and basal ganglia compared to blood [7]. In XDP patients, seven variants have been found on the X chromosome: five single-nucleotide variants (SNVs), a 48-bp deletion and the SVA insertion [8,9]. Within the SVA, no variants have been reported besides the (*CCCTCT*)_n_ repeat polymorphism [5].

The *TAF1* SVA insertion is also associated with decreased *TAF1* expression [6]. The reduced *TAF1* expression that has been observed in blood and patient-derived induced pluripotent stem cells can be rescued by excision of the retrotransposon insertion [10,11]. Thus, *TAF1* reduction is a consequence of the presence of the SVA insertion. However, how the *TAF1* SVA insertion influences gene expression levels still remains an enigma. Of note, two enhancers are predicted to be located upstream and ten enhancers downstream of the *TAF1* SVA insertion. The SVA itself is highly methylated due to the high “GC” content (~60%) within the variable number tandem repeat (VNTR) region, also known as “mobile CpG-island” [9,12]. Therefore, the SVA retrotransposon insertion may affect *TAF1* expression by changing the methylation status (causing hypo- or hypermethylation) of the surrounding genomic region across several enhancer sites. There are approximately 2700 SVA elements within the human reference genome (hg19) [13], and specific characterization of the *TAF1* SVA insertion in XDP patients has been hard to achieve with short-read sequencing technologies. *TAF1* SVA is a non-reference mobile element. Recently, mobile element insertions have been investigated in the context of the Simons Genome Diversity Project, and on average, 47 non-reference mobile element insertions are present per individual [14]. Similar to XDP, the insertion of SVAs have been implicated in many diseases such as neurofibromatosis type 1 and haemophilia B [15]. To our knowledge, the full-length *TAF1* SVA and flanking regions (>22 kb) have not been sequenced and investigated.

In this study, we establish a straightforward Nanopore sequencing workflow to investigate the genetic architecture of *TAF1* SVA by characterizing: (1) genomic variants within the SVA, (2) variations of the hexanucleotide repeat number and (3) CpG methylation by utilizing Nanopore long-read sequencing.

## 2. Materials and Methods

### 2.1. Patient Demographics

The study was approved by the Ethics Committees of the University of Luebeck, Germany and the Metropolitan Medical Center, Manila, Philippines (REF: IRB-MMC #: 10-073). For the analysis of genomic variants within the SVA and the detection of variations of the hexanucleotide repeat domain, *n* = 96 patients with XDP were investigated. As XDP follows an X-linked recessive inheritance pattern, only male patients were included. The mean age at onset (AAO) was 40.66 (SD = ±8.75), and the mean age at examination (AAE) and sample collection was 45.4 (SD = ±10.24) (Appendix A).

The CpG methylation was investigated in blood-derived DNA from one deceased XDP patient (L-7995) and one control (L-14529). The control was matched according to age, gender and ethnicity. For the patient (L-7995), brain tissue samples derived from the basal ganglia (BG) and cerebellum (CRB) were also available. The patient had an AAO of 31 years. The AAE was 36 years in the patient and the control.

### 2.2. Single-Nucleotide Variants and Repeat Detection

DNA was extracted with the Blood and Cell Culture DNA Midi kit (Qiagen). Long-range PCR was performed to amplify the *TAF1* SVA (amplicon Size: 3.2 kb) in XDP patients, as previously described [7], using the PrimeSTAR GXL DNA Polymerase^®^ (Takara Bio). The primer sequences are documented in Appendix A. Subsequently, 1 µg of each patient-derived PCR product was barcoded with the Native 96 Barcoding Kit (EXP-NBD196) and multiplexed. Two libraries with the Ligation Sequencing Kit (LSK109) were generated for Nanopore sequencing on two R9.4.1 flow cells on a GridION. The input for library preparation was 200 fmol of DNA per sample.

Validation by fragment analysis to determine the repeat length of the hexanucleotide (*CCCTCT*)_n_ was performed with a fluorescein amidites (FAM) labeled primer, as previously described [5,6].

### 2.3. Methylation Detection

To obtain the epigenetic information and to enrich the target region, Cas9-targeted sequencing from Oxford Nanopore Technology (ONT) was performed. For the specific ligation of the sequencing adapter, the blunt ends with 5′ phosphates resulting from the Cas9 ribonucleoprotein complex, cleaving out the region of interest. The CRISPR RNAs (crRNAs) were designed with the ChopChop tool (https://chopchop.cbu.uib.no, accessed on 10 December 2021) [16]. Four crRNAs were used upstream of the *TAF1* SVA insertion, and four crRNAs were used downstream (Appendix A). Two different library preparations were used for the Cas9-targeted enrichment. The first library consisted of the full ~22 kb region of interest (crRNA 1, 2, 7 and 8). The second library targeted a 5.5 kb product specifically around the SVA (crRNA 3, 4, 5 and 6). For the control without an SVA insertion, the second target was 2.8 kb in size. We prepared multiple libraries for the DNA derived from one patient (L-7995) or one control (L-14529). To prepare the individual libraries, two crRNAs were used to cut upstream of the target region and two downstream to enhance the efficiency of Cas9 DNA cleavage. For the blood-derived DNA of the patient with XDP, we have used five flow cells (R9.4.1) loaded with six libraries (5 × 5 µg and 1 × 1 µg). For the BG-derived DNA of the patient with XDP, we have used five flow cells (R9.4.1) loaded with seven libraries (2 × 5 µg, 3 × 3 µg, 1 × 2 µg and 1 × 1 µg). For the CRB-derived DNA of the patient with XDP, we have used four flow cells (R9.4.1) loaded with eight libraries (7 × 5 µg and 1 × 1 µg). For the blood-derived DNA of the healthy control, we have used five flow cells (R9.4.1) loaded with six libraries (5 × 5 µg and 1 × 1 µg).

The enriched DNA was prepared with the Nanopore Ligation Sequencing Kit (SQK-LSK109), loaded on an R9.4.1 flow cell and sequenced with the MinION or GridION. For methylation analysis, all sequencing data obtained were combined to maximize coverage depth.

### 2.4. Data Analysis

Base-calling was performed with Guppy version 5.0.11, and the base-calling software is available for Nanopore community members (https://community.nanoporetech.com, accessed on 10 December 2021). For the detection of the repeat length, the super accuracy model (DNA_r9.4.1_450bps_sup.cfg) and the fast model (DNA_r9.4.1_450bps_fast.cfg) were used. The corresponding configuration file names were provided as a parameter to the Guppy software. The expected base-calling accuracy for the super accuracy model is 98.3% and 95.8% for the fast model. (https://community.nanoporetech.com/posts/guppy-v5-0-7-release-note, accessed on 10 December 2021). Base-calling for the methylation detection was performed with the fast model. All reads were mapped to the reference sequence with the software Minimap2 (v2.17). The coverage was determined with the software Samtools (v1.9).

Variants were identified with the software Bcftools (v1.9) (https://github.com/samtools/bcftools, accessed on 10 December 2021). To prevent false-positive results, all reported positions by Bcftools were controlled in the VCF file. We filtered for hemizygous allelic frequency (>90%) and good quality (Phred score Q > 20). Lastly, variants were evaluated in the Integrative Genomics Viewer (IGV) to exclude erroneously called variants within homopolymeric stretches.

Detection of the repeat length was conducted using the NCRF software (Noise-canceling repeat finder) (v1.01.02) [17]. To determine the repeat length for one patient, the median of all reads was calculated as previously described [7]. The NCRF alignment was used additionally to explore the frequency of deletions, insertions and mismatches within the repeat motif.

For the Cas9-targeted sequencing data, methylation was called using the software Nanopolish (v0.13.2), which can detect 5′-methylcytosine (5 mC) in a CpG context. Nanopolish requires, besides the FASTQ and FAST5 files, the alignment in a BAM format as an input. To counteract potential off-target effects of the CRISPR-Cas9 enrichment, the BAM file was filtered for reads with an alignment length >3 kb in the patient- or >1.5 kb in control-derived samples. Only CpG sites covered by >10 reads were included in the analysis.

### 2.5. Statistical Analysis

Spearman correlation was performed to assess the concordance of the detected hexanucleotide repeat number between Nanopore sequencing and fragment analysis. The median repeat number and the interquartile range detected by NCRF from the Nanopore data, and the number of repeats detected with fragment analysis were used for the correlation plot.

In addition, we used the NCRF report for each sample to assess repeat motif interruptions. To determine matches and mismatches (i.e., deletions, insertions and substitutions) between the Nanopore reads and the hexanucleotide repeat motif, NCRF uses a Smith–Waterman aligner approach and affine gap penalties [17]. The software reports the total number of deletions, insertions and substitutions per read. Subsequently, the mean number of repeat motif interruptions per read across all samples were calculated, as reported by NCRF, to explore accumulations of deletions, insertions and substitutions within the SVA hexanucleotide repeat domain.

DNA methylation was compared across different tissues of a patient with XDP and a control. These differences were assessed by a non-parametric Mann–Whitney U-test, as previously described in Ewing et al. 2020 [12].

## 3. Results

We first analyzed the sequencing data generated by PCR amplification and subsequent multiplexing on the Nanopore of the *TAF1* SVA insertion (*n* = 96 XDP patients). Across all individuals, we obtained a mean coverage of 17,645X (SD = ±12,392X) per barcode. The mean coverage of the samples ranged between 1690X (SD = ±190X) and 47,919X (SD = ±5074X) due to the variable sequencing efficiency of the barcodes. However, the coverage of the amplified region was even within the samples (Appendix A). The mean sequence quality (Phred score) was 15.88 (SD = ±0.44), and the mean N50 was 3.38 kb (SD = ±66.42 bp) per barcoded sample.

### 3.1. Single-Nucleotide Polymorphisms within the SVA TAF1 Insertion

SNVs located within the *TAF1* SVA insertion were called from the amplicon sequencing data of all 96 patients. After quality filtering and the final evaluation with IGV, no SNVs were detected.

### 3.2. Assessment of the Hexanucleotide Repeat Length

The hexanucleotide repeat number detection with long-read sequencing amplicon data resulted in a mean of 45.17 (SD = ±4.24) repeats, ranging from 35 to 57, using super accuracy base-calling (Figure 1A). Fragment analysis as an independent validation showed a mean number of 42.21 (SD = ±4.23) repeats, ranging from 33 to 54. The detected repeat numbers were highly concordant between the two methods (Spearman’s *r* = 0.9765, Spearman’s exploratory *p*-value < 1 × 10^−15^, Figure 1A). However, the repeat number detected from long-read sequencing was consistently 1–4 repeat numbers higher compared to fragment analysis. Using Guppy fast base-calling for Nanopore sequencing resulted in a mean of 42.77 (SD = ±4.05) repeats, ranging from 33 to 54. Thus, we observed a higher concordance with fast base-calling between both methods (Spearman’s *r* = 0.9883, Spearman’s exploratory *p*-value < 1 × 10^−15^, Figure 1B). There was an identical repeat number in *n* = 47 patients and a difference of ~1–2 repeats in *n* = 48 patients.

To further validate our workflow, we analyzed the previously shown negative association between the AAO and the repeat number [5,6]. The repeat number detected with Nanopore sequencing negatively correlated with AAO in patients with XDP (Spearman’s *r* < −0.80, Spearman’s exploratory *p*-value < 1 × 10^−15^).

Next, we explored the continuity of the repeat motif. As reported by NCRF, the mean number of deletions per read within the repetitive sequence was 6.05 (SD = ±1.00), the mean number of insertions was 2.89 (SD = ±0.36) and of substitutions 0.73 (SD = ±0.09). Thus, deletions were the most common type of interruptions detected in the hexanucleotide repeat sequence of XDP patients (Figure 2).

### 3.3. Methylation within the SVA and in the Flanking Regions

To assess the DNA CpG methylation of the SVA, we enriched the *TAF1* SVA insertion and flanking regions with a Cas9-targeted approach. We included blood- and brain-derived DNA from one XDP patient and blood-derived DNA from one age-matched control participant. We used two Cas9 enrichment strategies: (1) the *TAF1* SVA insertion and a short flanking region (~5.5 kb) and (2) the *TAF1* SVA insertion and a longer flanking region (~22 kb), including 12 predicted enhancer sites.

The enrichment of the shorter fragment resulted in an N50 of 4.5 kb for the patient-derived samples and an N50 on 2.0 kb for the control-derived sample. The mean Phred score of the reads ranged from 10.0 to 10.9 and the mean coverage from 126.9X (SD = ±79.8X) to 1226.0X (SD = ±554.9X) (Appendix A).

The enrichment of the longer fragment resulted in an N50 of 4.7 kb for the patient-derived samples and an N50 on 8.8 kb for the control-derived sample. The mean Phred score of the reads ranged from 12.3 to 13.5 and the mean coverage from 22.1X (SD = ±11.5X) to 591.0X (SD = ±1202.0X). The sequencing quality statistics were summarized in Appendix A.

Overall, the methylation levels within the SVA as well as in the up- and downstream flanking regions were high in the patient-derived samples (Figure 3A). However, the mean MF was lower in the brain-derived samples (BG: mean MF ± SD = 0.88 ± 0.15, CRB: mean MF ± SD = 0.90 ± 0.14) compared to the blood-derived sample (mean MF ± SD = 0.94 ± 0.12). There were *n* = 153 CpG sites within the SVA *TAF1* insertion (Figure 3B). Consistent with the overall methylation level across the 22 kb region, the mean CpG MF within the SVA specifically was still lower in the brain-derived samples (BG: mean MF ± SD = 0.87 ± 0.14, CRB: mean MF ± SD = 0.93 ± 0.08) compared to the blood-derived sample (mean MF ± SD = 0.96 ± 0.07) (exploratory Mann–Whitney U-test *p* < 1.2 × 10^−6^) (Appendix A). In addition to patient-derived DNA, we analyzed blood-derived DNA from one healthy control (Figure 3C). The overall MF across the SVA flanking region in the control sample was at 0.83 ± 0.17, which was lower than the patient-derived sample (MF ± SD = 0.93 ± 0.15) (exploratory Mann–Whitney U-test *p* < 1 × 10^−15^, Appendix A). Despite a significant difference, the effect size was small.

There were 12 predicted enhancer sites located in the targeted region, 2 upstream and 10 downstream of the *TAF1* SVA insertion, according to the ENCODE project (reference number: wgEncodeEH000790). There was no CpG site located within enhancer eight, and this predicted enhancer was excluded from the analysis. The mean MF of the enhancer sites ranged from 0.65 to 0.99 in the blood-derived sample, from 0.46 to 0.99 in the BG-derived sample and from 0.37 to 0.99 in the CRB-derived sample (Figure 4A, Appendix A). In comparison, the mean MF of these enhancers ranged from 0.69 to 0.95 in the healthy control (Figure 4B, Appendix A). We detected significantly lower methylation of the enhancer sites within the *TAF1* SVA flanking region in the control compared to the patient-derived blood sample (exploratory Mann–Whitney U-test *p* = 0.0033, Appendix A). With the exception of enhancer two, all enhancers showed lower methylation levels in the control subject. The most pronounced difference was observed at enhancer site six (mean MF patient: 0.91, mean MF control: 0.71) and nine (mean MF patient: 0.98, mean MF control: 0.70). However, the sample size is small, the effect sizes are small, and differences remain difficult to interpret (see details in Section 4).

## 4. Discussion

In this study, we performed Nanopore single-molecule sequencing to examine the genetics and epigenetics of the full-length *TAF1* SVA insertion in patients with XDP. The novelty of our study lies in: (1) a new multiplexed workflow to quantify the *TAF1* SVA repeat number in patients that shows high concordance with fragment analysis which can be used as a cost-effective diagnostic tool in the future; (2) the exploration of novel variants within the SVA besides the repeat motif across 96 patients which has not been possible with older technologies and lastly; (3) the detection of direct CpG methylation across the full-length SVA and flanking regions up to 22 kb which incorporates 12 putative enhancer sites.

### 4.1. Examination of the Hexanucleotide Repeat Domain

Deep Nanopore sequencing (>5000X) of the GC-rich *TAF1* SVA insertion allowed better alignment accuracy for the low-complexity repetitive regions of the SVA than short-read sequencing approaches [18]. From our sequencing data analysis, there was no evidence of any other genetic variability besides the repeat domain within the investigated locus. Thus, variability of AAO and disease severity largely result from the hexanucleotide repeat number within the SVA [5,6]. As the sequence of the SVA is identical between patients except for the polymorphic repeat, this further validates the notion that the insertion of this repeat sequence in intron 32 of *TAF1* causes XDP [19].

The software tool, NCRF, has been specifically designed to explore repetitive sequences in noisy long-read sequencing data [17]. More specifically, the noise in the Nanopore signal trace is due to indel and homopolymer errors, reducing sequencing accuracy [17]. To decrease the noise even further, we performed base-calling with the novel “super-high accuracy” model with improved read accuracy provided by the Nanopore software Guppy (v.5.0.11). Indeed, the repeat number resulting from the NCRF analysis was highly concordant with the results from the independent fragment analysis method. In concordance with the literature [5,6], the repeat number detected by Nanopore sequencing was negatively associated with the AAO of patients with XDP as well, which further validates our workflow. Interestingly, the detected repeat number by NCRF was consistently 1–4 repeats longer than the number obtained from the fragment analysis. The slightly larger repeat number, detected with Nanopore sequencing, could be due to deviations in the repeat pattern. We did not detect a consistently higher repeat number when we compared the results from the fast base-calling to the fragment analysis. The higher concordance with the repeat number detected with fast base-calling could result from general repeat number detection of fragment analysis without the resolution of mismatches in the repeat motif. In fact, there was a noticeable increase in the frequency of deletions in the long-read data that require further exploration as interruption of the repeat motif has also been reported for other repeat expansion diseases such as Friedreich’s Ataxia or Huntington disease [20,21]. Therefore, further investigation of this issue, including single nucleotide resolution of the repeat interruption, is required.

### 4.2. Methylation Status of the TAF1 SVA Insertion and Adjacent Enhancer Sites

Nanopore sequencing has been used to investigate DNA methylation in the context of other repeat expansions before [12,22]. Recently, in the context of cancer, Nanopore sequencing has been used to assess the genetics and epigenetics of transposable elements simultaneously, including the CG-rich VNTR domain of SVAs, also known as “mobile CpG-island” [12]. To maintain DNA methylation, an amplification-free Cas9-guided approach for Nanopore sequencing was introduced [23]. This targeted approach has been shown to efficiently enrich repeat expansions causing frontotemporal dementia and amyotrophic lateral sclerosis or fragile X syndrome [22].

In this study, the target region was enriched against the genomic background DNA, and methylation was maintained, using a Cas9-guided approach. Coverage of the SVA insertion specifically was high; however, lower sequencing depth was obtained in the flanking regions. The variability in coverage could be due to the limitation of different targeting efficiencies of the crRNAs used in the Cas9-approach. To counteract potential off-target effects, we included only reads with sufficient alignment length to the reference sequence.

We observed hypermethylation of the *TAF1* SVA insertion in concordance with the literature [9]. Although the overall MF in all patient-derived samples was high, it was mildly reduced in the brain-derived samples. There have been speculations that neuron-specific expression of *TAF1* is reduced in patients with XDP [9]. However, recently published studies could not confirm a significant decrease of neuron-specific *TAF1* in XDP patients [24,25]. In addition, it is unclear if the slight change of the methylation level in the brain-derived samples that we observed could affect the expression level of *TAF1* and whether it would be relevant for the disease to develop. Furthermore, we did not specifically analyze the methylation level of neurons, which could be a perspective for future experiments.

There is the possibility that retrotransposon insertions introduce methylation changes into the flanking regions [26]. There have been speculations that the hypermethylation of the SVA could also affect the methylation of adjacent *cis*-regulatory elements [9]. The 12 predicted enhancer sites in the target region showed mostly comparable methylation levels in the blood and brain-derived samples, and only 2 enhancer sites showed a slight methylation decrease in the brain. Interestingly, lower methylation levels of the BG-derived sample, in particular, were present in the SVA insertion, as well (Figure 3B). This difference is most pronounced in the VNTR domain of the retrotransposon. The lower methylation observed in the BG can be explained by tissue-specific differences and is not necessarily a disease-related phenomenon. Indeed, tissue-specific methylation patterns of transposable elements have been investigated with Nanopore sequencing [12,27]. Of particular interest is the reduced methylation of intergenic subfamily SVA_f_ insertion in the X-chromosome in tumor tissues compared to non-tumor tissues [12]. In general, overall CpG methylation of brain tissue and peripheral tissues is highly correlated across participants. On the other hand, differences between the methylation level of the brain and peripheral tissues within a particular individual are possible, which can contribute to tissue-specific gene expression [28,29,30].

The goal of this study was to establish a new workflow for direct CpG methylation detection using Nanopore sequencing. The study shows that it is indeed possible to detect methylation across a large region, including the *TAF1* SVA of 22 kb. There has been evidence that DNA methylation can be a molecular mechanism in XDP pathogenesis. Due to the abolishment or introduction of CpG dinucleotides by disease-specific single-nucleotide changes (DSC), significant differences in methylation between XDP patients and controls at these positions have been reported, suggesting a potential effect on *TAF1* expression [31]. These three reported DSCs are located ~700 kb away from the SVA. Our study focused on regions within and adjacent to the SVA insertion. As the GC-content of the VNTR is high and the SVA is known to be hypermethylated, the change in the direct genetic environment could lead to altered methylation patterns within the SVA and flanking regions. As a pilot study, we observed overall lower methylation of the control sample compared to the XDP patient in the SVA flanking region as well as in 10 out of the 12 predicted enhancer sites. The observed methylation differences in this study are only preliminary and should be interpreted with caution. Another limitation of this study was the lack of brain-derived DNA from controls. Thus, more individuals and tissue types should be investigated using the Cas9-targeted approach across this 22 kb region. Still, our results show the utility of long reads in detecting the full-length SVA in *TAF1* in the context of XDP.

## 5. Conclusions

In this study, we present a straightforward and scalable long-read deep sequencing workflow to quantify the hexanucleotide repeat number of the *TAF1* SVA in patients with XDP. The high concordance of the results obtained from Nanopore sequencing to independent fragment analysis highlights the accuracy of our workflow. The long reads were also utilized to investigate variations within the SVA locus other than the repeat motif. As the sequence of the SVA locus was identical between patients besides the hexanucleotide repeat domain, our results further underline that the insertion of this repeat sequence is associated with the variability in AAO and expressivity in XDP. Lastly, an amplification-free Cas9 targeted enrichment of the SVA locus and the flanking regions allowed us to comprehensively assess the (epi-) genetics of the *TAF1* SVA locus.

## Figures and Tables

**Figure 1 genes-13-00126-f001:**
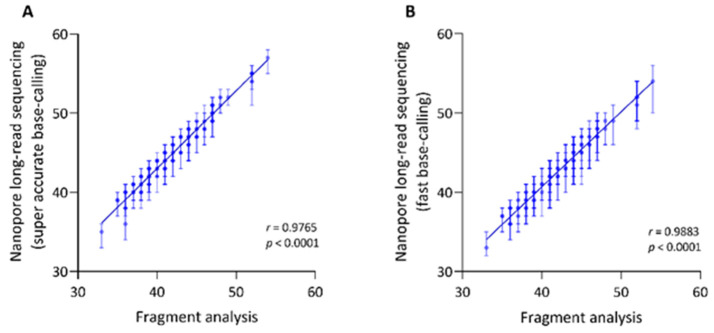
Repeat number detection using Nanopore long-read sequencing is highly concordant with the results from fragment analysis. Correlation between the median repeat numbers per individual of the (*CCCTCT*)_n_ SVA domain, detected with fragment analysis and Nanopore sequencing using super accurate (**A**) or fast base-calling (**B**). Bars indicate the interquartile range of the detected repeat number with Nanopore sequencing. *R* = Spearman’s rank correlation coefficient, *p* = Spearman’s exploratory *p*-value.

**Figure 2 genes-13-00126-f002:**
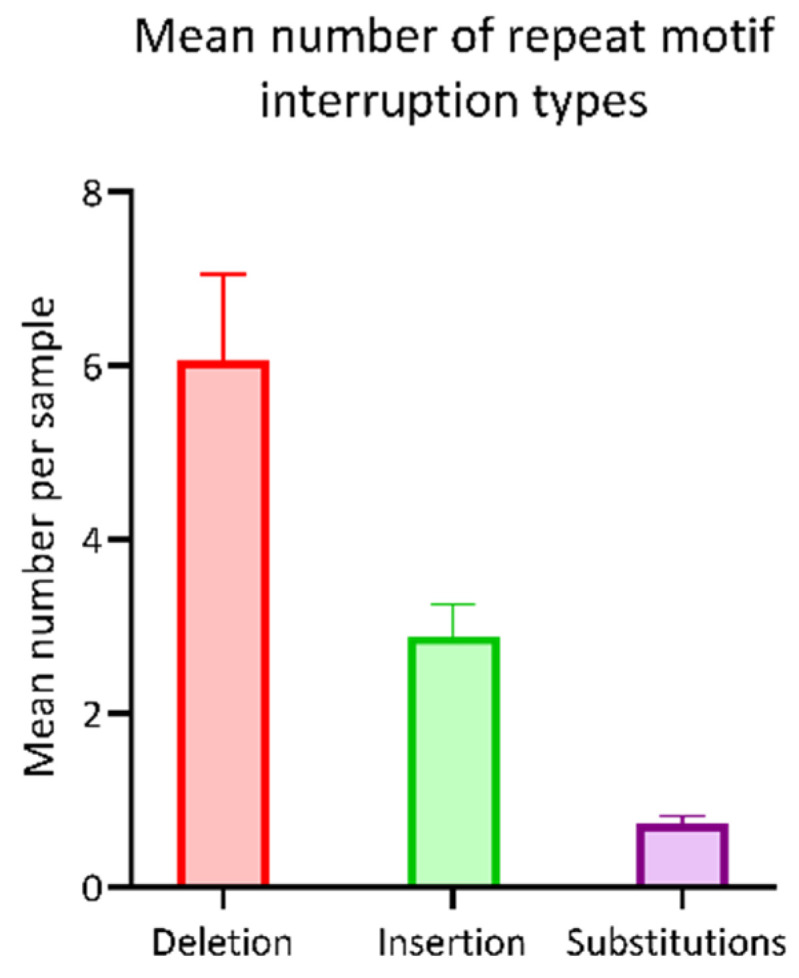
Occurrence of repeat motif interruptions. The bar chart shows the mean number of repeat motif interruptions per patient sample, stratified by type (i.e., deletion, insertion, substitution). The bars and whiskers represent the mean and upper limit of the standard deviation.

**Figure 3 genes-13-00126-f003:**
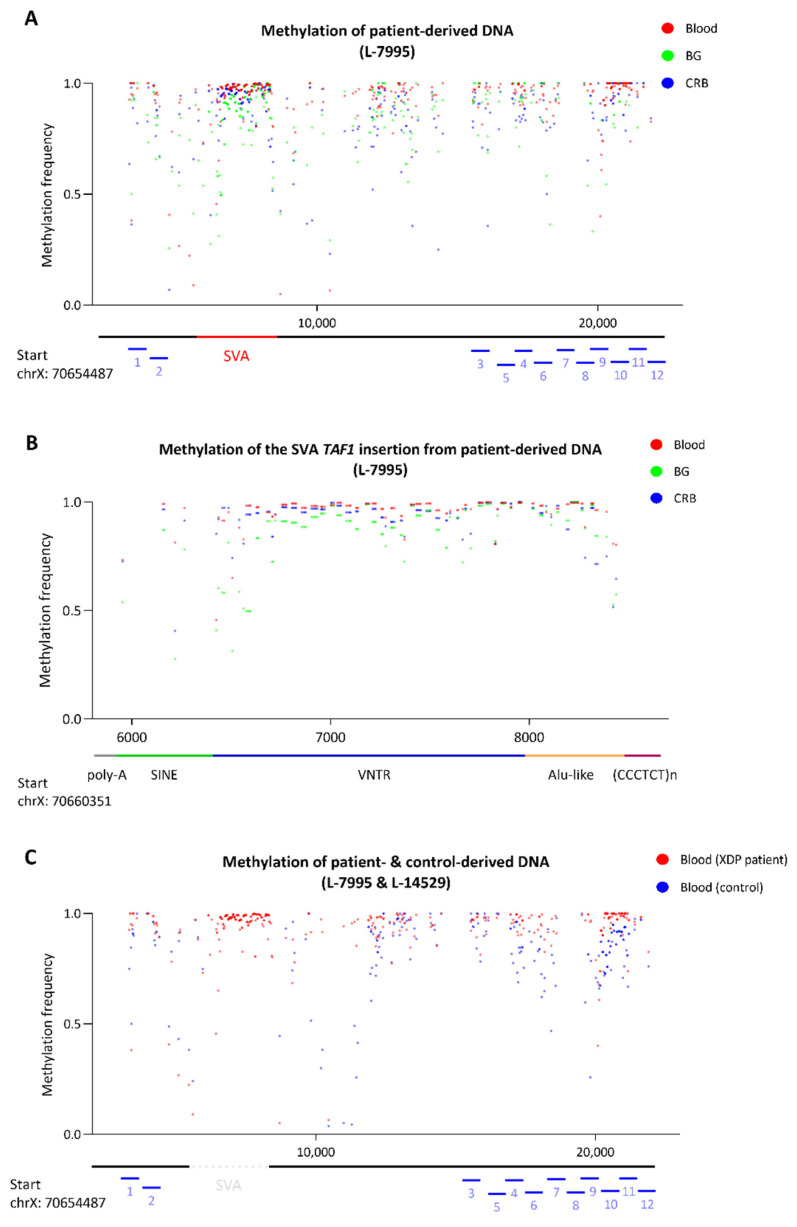
Methylation frequency of the *TAF1* SVA insertion and flanking regions. (**A**) Methylation frequency of two different brain tissues and blood-derived from a patient with XDP. Red indicates methylation from blood-derived DNA, green from basal ganglia-derived (BG) DNA and blue from cerebellum-derived (CRB) DNA. (**B**) Methylation frequency of *TAF1* SVA insertion with indicated SVA domains of the same patient-derived DNA samples. Red indicates methylation from blood-derived DNA, green from BG-derived DNA and blue from CRB-derived DNA. (**C**) Methylation frequency of blood-derived DNA from a patient with XDP (red) and a control participant (blue). The *x*-axis indicates the position in the reference sequence. The bars indicate the location of predicted enhancers, the *TAF1* SVA insertion and the insertion’s subunits.

**Figure 4 genes-13-00126-f004:**
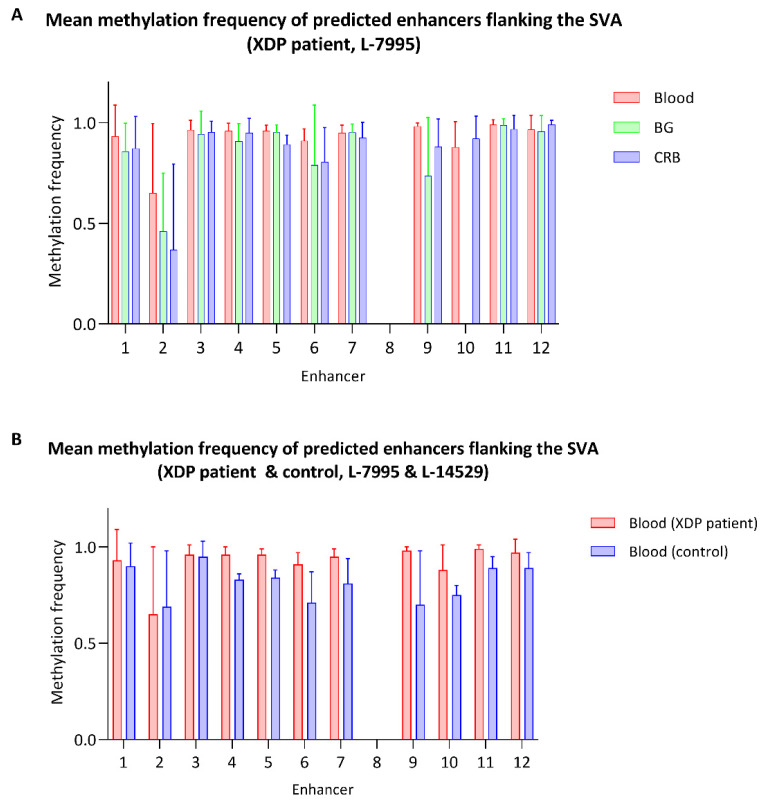
Methylation levels of the predicted enhancers flanking the *TAF1* SVA. (**A**) The bar plot shows the CpG methylation frequency of predicted enhancer sites within the targeted region. DNA was derived from the blood, basal ganglia and cerebellum of a XDP patient (**A**) or derived from the blood of a patient and a healthy control participant (**B**). The bars and whiskers represent the mean and upper limit of the standard deviation of the methylation frequency from the CpG sites within a predicted enhancer.

## Data Availability

The data presented in this study are available on SRA (SAMN24775867-SAMN24775962, SAMN24115523-SAMN24115530). The bioinformatical commands to quantify the *TAF1* SVA (*CCCTCT*)_n_ repeat length are described here: https://github.com/nanopol/xdp_sva/ (accessed on 10 December 2021).

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
