# Peer review of "Elucidating Hexanucleotide Repeat Number and Methylation within the X-Linked Dystonia-Parkinsonism (XDP)-Related SVA Retrotransposon in TAF1 with Nanopore Sequencing"

_genes, 2022, doi:10.3390/genes13010126_

Round 1

Reviewer 1 Report

The authors answered all my questions.

Author Response

Comments and Suggestions for Authors:
The authors answered all my questions.
We thank the Reviewer for the careful and positive review of our manuscript and we are pleased that we were able to answer all the questions.

Reviewer 2 Report

This manuscript presented novel findings on the genome repeat as genetic modifier of X-linked dystonia-parkinsonism disease based on Nanopore sequencing.

The material is well-balanced, fits to the journal scope.

There are interesting fundamental problems on the sequencing technology limits raised.

I have only some minor remarks to fix:

In the Abstract - may not use (1)(2)(3)... numbering before ‘Background’, ‘Methods’ etc.  words starting corresponding subsections of the Abstract.

Line 41: The sentence ‘Herein we use Nanopore..’ could be in the ‘Methods’ subsection of the Abstract, since it is about technology.

Line 44: ‘it  with fragment  analysis’ - could add details here about the fragments analysis (name the technology).

Line 47: ‘no difference in genetic variability’ - I’d add wording about no difference in the locus studied. The difference might be in other gene regions that were not sequenced.

Line 50: ‘...flanking the SVA...’ - add word ‘SVA region’ or ‘SVA locus’ to be precise.

Line 52: ‘may not lead..’ - This is not proper conclusion. It was not detected, but may lead? Please rephrase. Or the insertion itself doesn’t affect the disease, but the hexanucleotide repeat length correlate to the disease onset.

Line 59: ‘mainly from the Philippines or are of Filipino descent’ - what is ‘mainly’? May add details (like 90% or 100%?)

Are any data on genetic backgrounds (common ancestor) of the patients? Just add a phrase here or in the Discussion.

Line 64: ‘(CCCTCT)n’ - letter ‘n’ is not in proper font. (see also line 71)

Line 79: ‘VNTR region’ - the abbreviation VNTR should be given in full here.

Line 82: ‘2700 SVA elements within  the human genome [12]’ - the reference [12] is old. What is the number from recent human genome assembly? Is any association of SVA elements to other genetic diseases know?

Line 92: ‘study was approved’ - please add a document number to refer or add a phrase like ‘(see Supplement)’

Line 114: ‘FAM-tagged’ - give abbreviation FAM in full.

Line 119: ‘crRNAs’ - give full name of crRNA

Line 120: ‘ChopChop’ - add word ‘tool’, give a reference to the software.

Line 128: ‘the patient with XDP’ - it is not clear if it is single patient, or 96 patients. Please add a phrase. Several libraries were prepared from single patients data?

Line 136: ‘MinION or GridION’ - how many samples were sequenced by CridION? Why word ‘or’? IS any difference in datasets obtained from these instruments?

Line 140: ‘Guppy version 5.0.11’ - need a reference (might be online link)

Line 141: ‘super  accuracy  model’ - it is not clear what is the model. Configuration file names also not understandable for reader. Please comment on the accuracy and fast models, add references or describe. What is ‘(dna_r9.4.1_450bps_sup.cfg)’?

Line 146: ‘Bcftools’ - give the reference or link to this tool.

Line 149: ‘IGV’ - give the abbreviation in full.

Line 151: ‘software “Noise-canceling repeat finder” (NCRF)’ - it is correct naming and citation, but I’d recommend write full name after the abbreviation:

‘NCRF software (Noise-canceling repeat finder)’ to avoid extra parentheses.

Line 158: ‘Fastq’ - should be in capital - FASTQ.

Line 166: ‘IQR’ - give the abbreviation in full

Line 168: ‘..NCRF report per sample..’ - it means per each sample? Or only one sample?

Line 223: ‘bars represent..’ - could be ‘bars and whiskers’ words.  And the standard deviation is not visible in the figure (only top part f thin lines visible)

Line 255: ‘ENCODE project’ - add reference, the database release number.

Line 265: ‘six. (’    - extra dot here. Why not wrote by digits - 6?

Line 267: ‘the effect sizes are small and differences remain difficult to interpret.’ - in fact here was only one comparison between the patient and control, but using several tissues and libraries. Is it correct? May add a phrase here: ‘(see details in Discussion section)’.

Line 278: ‘x-axis’ - may change to ‘axis X’

Line 300: ‘any other genetic variability besides the repeat domain’ - may add details her - it is only about X chromosome. The variability could be in other region, not sequenced around TAF1 gene.

Line 313: ‘These results obtained..’ - the phrase is not correct. The authors obtained new results by Nanopore that confirm previously published results in [4,5]. The sentence should be rewritten.

Line 323: ‘other repeat expansion diseases...’  - I’d recommend name these diseases, i.e. Friedreich's Ataxia, not just word ‘other’.

Line 330: ‘mobile CpG-island’ - this is good comparison, may use this wording earlier in the text, in Introduction section.

Lines 345-346: (N-TAF1) - this abbreviation used only once, may remove it for better readability of the text.

Line 362: ‘SVAf ‘ - what is lower case letter ‘f’, a typo?

Line 379: ‘As a pilot, we..’ - use wording ‘pilot study’, not just ‘pilot’.

Line 388: ‘.,  title, Table’ - some redundant dots and word ‘title’.

 Avoid ‘.,’ in the text, use ‘;’ sign instead.

Author Response

Comments and Suggestions for Authors
This manuscript presented novel findings on the genome repeat as genetic modifier of X-linked dystonia-parkinsonism disease based on Nanopore sequencing.
The material is well-balanced, fits to the journal scope.
There are interesting fundamental problems on the sequencing technology limits raised.
Response: We thank the Reviewer for their appraisal of this work and very detailed review that further strengthens our paper.

I have only some minor remarks to fix:
In the Abstract - may not use (1)(2)(3)... numbering before ‘Background’, ‘Methods’ etc. words starting corresponding subsections of the Abstract.
Line 41: The sentence ‘Herein we use Nanopore..’ could be in the ‘Methods’ subsection of the Abstract, since it is about technology.
Response: We have now adjusted the Abstract to incorporate both changes.

Line 44: ‘it with fragment analysis’ - could add details here about the fragments analysis (name the technology).
Response: We have now included specifics regarding the fragment analysis.
Abstract: “We used blood-derived DNA from 96 XDP patients for amplicon-based deep Nanopore sequencing and validated it with fragment analysis which was performed using fluorescence-based PCR.”

Line 47: ‘no difference in genetic variability’ - I’d add wording about no difference in the locus studied.
The difference might be in other gene regions that were not sequenced.
Line 50: ‘...flanking the SVA...’ - add word ‘SVA region’ or ‘SVA locus’ to be precise.
Response: We have now revised the Abstract to specify the region or ‘locus’ and used ‘SVA locus’ throughout.
Abstract: “Within the SVA locus, there was no difference in genetic variability other than variations of the repeat motif between patients.”

Line 52: ‘may not lead..’ - This is not proper conclusion. It was not detected, but may lead? Please rephrase. Or the insertion itself doesn’t affect the disease, but the hexanucleotide repeat length correlate to the disease onset.
Response: We have shortened and adjusted the conclusion.
Abstract: “Nanopore sequencing can reliably detect SVA hexanucleotide repeat numbers, methylation and lastly, variation in the repeat motif.”

Line 59: ‘mainly from the Philippines or are of Filipino descent’ - what is ‘mainly’? May add details (like 90% or 100%?)
Response: Patients originate mainly from the Philippines or are of Filipino descent and mainly aggregate in the island of Panay. Unfortunately, the exact percentage of individuals who are of Filipino descent but are not born in the Philippines has not been estimated in the literature. 

Are any data on genetic backgrounds (common ancestor) of the patients? Just add a phrase here or in the Discussion.
Response: Indeed, the genetic cause of XDP originated from a founder mutation.
“A known family history of the disease is present for ~94% of the patients. XDP originated through a founder mutation approximately 1000 years ago.”

Line 64: ‘(CCCTCT)n’ - letter ‘n’ is not in proper font. (see also line 71)
Line 79: ‘VNTR region’ - the abbreviation VNTR should be given in full here.
Response: We have corrected the font for repeat motif and have added the full name for VNTR.

Line 82: ‘2700 SVA elements within the human genome [12]’ - the reference [12] is old. What is the number from recent human genome assembly? Is any association of SVA elements to other genetic diseases know?
Response: We acknowledge that the reference is older and only investigates the reference genome and realize that a more recent citation from 2020 (the Simons Genome Diversity Project) which encompasses non-reference mobile elements are even more relevant to the TAF1 SVA insertion (considering that this a non-reference mobile element itself). Lastly, the insertion of SVAs at other genetic loci is also associated with diseases such as Neurofibromatosis type 1 and haemophilia B (Pfaff, A.L. et al. 2021).
Introduction:
“There are approximately 2700 SVA elements within the human reference genome (hg19) and specific characterization of the TAF1 SVA insertion in XDP patients has been hard to achieve with short-read sequencing technologies. TAF1 SVA is a non-reference mobile element. Recently, mobile element insertions have been investigated in the context of the Simons Genome Diversity Project, and on average, 47 non-reference mobile element insertions are present per individual (Watkins, S.W. et al. 2020). Similar to XDP, insertion of SVAs have been implicated in many diseases such as Neurofibromatosis type 1 and haemophilia B (Pfaff, A.L. et al. 2021)”

Line 92: ‘study was approved’ - please add a document number to refer or add a phrase like ‘(see Supplement)’
Response: We have added the reference number of the ethics approval to the Methods section.

Line 114: ‘FAM-tagged’ - give abbreviation FAM in full.
Line 119: ‘crRNAs’ - give full name of crRNA
Response: We added the full name for all abbreviations.

Line 120: ‘ChopChop’ - add word ‘tool’, give a reference to the software.
Response: We added the word ‘tool’ and corresponding reference to the Methods section.

Line 128: ‘the patient with XDP’ - it is not clear if it is single patient, or 96 patients. Please add a phrase.
Several libraries were prepared from single patients data?

Line 136: ‘MinION or GridION’ - how many samples were sequenced by GridION? Why word ‘or’? IS any difference in datasets obtained from these instruments?
Response: There is no difference between these platforms as we use the same software and the same flow-cells to perform Nanopore sequencing with either the MinION or the GridION device.

Line 140: ‘Guppy version 5.0.11’ - need a reference (might be online link)
Response: The Guppy base-calling software is available for Nanopore community members (https://community.nanoporetech.com) and we have added the information to the Methods section.

Line 141: ‘super accuracy model’ - it is not clear what is the model. Configuration file names also not understandable for reader. Please comment on the accuracy and fast models, add references or describe. What is ‘(dna_r9.4.1_450bps_sup.cfg)’?
Response: We agree with the reviewer and expanded the explanation about the base-calling models.
Methods: “Base-calling was performed with Guppy version 5.0.11 and the base-calling software is available for Nanopore community members (https://community.nanoporetech.com). For the detection of the repeat length, the super accuracy model (dna_r9.4.1_450bps_sup.cfg) and the fast model (dna_r9.4.1_450bps_fast.cfg) were used. The corresponding configuration file names were provided as a parameter to the Guppy software. The expected base-calling accuracy for the super accuracy model is 98.3% and 95.8% for the fast model. (https://community.nanoporetech.com/posts/guppy-v5-0-7-release-note).”

Line 146: ‘Bcftools’ - give the reference or link to this tool.
Response: We have added the link to this tool.

Line 149: ‘IGV’ - give the abbreviation in full.
Response: We have added the full name to the abbreviation.

Line 151: ‘software “Noise-canceling repeat finder” (NCRF)’ - it is correct naming and citation, but I’d recommend write full name after the abbreviation:
‘NCRF software (Noise-canceling repeat finder)’ to avoid extra parentheses.
Line 158: ‘Fastq’ - should be in capital - FASTQ.
Line 166: ‘IQR’ - give the abbreviation in full
Response: We added the full name for the abbreviations and modified the texts to incorporate changes.

Line 168: ‘..NCRF report per sample..’ - it means per each sample? Or only one sample?
Response: We thank the reviewer for raising this point. It means for each of the 96 samples. We have changed the text accordingly.

Line 223: ‘bars represent..’ - could be ‘bars and whiskers’ words. And the standard deviation is not visible in the figure (only top part f thin lines visible)
Response: We have changed the text accordingly and the legend now reads “bars and whiskers represent the mean and upper limit of the standard deviation”.

Line 255: ‘ENCODE project’ - add reference, the database release number.
Response: We have included the ENCODE reference number (wgEncodeEH000790).

Line 265: ‘six. (’ - extra dot here. Why not wrote by digits - 6?
Response: We have removed the dot here and kept the digit as written form.

Line 267: ‘the effect sizes are small and differences remain difficult to interpret.’ - in fact here was only one comparison between the patient and control, but using several tissues and libraries. Is it correct?
May add a phrase here: ‘(see details in Discussion section)’.
Response: That is correct, we have included the phrase.

Line 278: ‘x-axis’ - may change to ‘axis X’
Response: We refer to the horizontal plane of the graph. Thus, we used the algebraic nomenclature.

Line 300: ‘any other genetic variability besides the repeat domain’ - may add details her - it is only about X chromosome. The variability could be in other region, not sequenced around TAF1 gene.
Response: Please see above comment regarding the locus, we have now consistently modified this in the manuscript.
Discussion: “From our sequencing data analysis, there was no evidence of any other genetic variability besides the repeat domain, within the investigated locus.”

Line 313: ‘These results obtained..’ - the phrase is not correct. The authors obtained new results by Nanopore that confirm previously published results in [4,5]. The sentence should be rewritten.
Response: We have changed the text accordingly.
Discussion: “In concordance with literature, the repeat number detected by Nanopore sequencing was negatively associated with the AAO of patients with XDP as well, which further validates our workflow.”
Line 323: ‘other repeat expansion diseases...’ - I’d recommend name these diseases, i.e. Friedreich's Ataxia, not just word ‘other’.
Response: We have now included “In fact, there was a noticeable increase in the frequency of deletions in the long-read data that require further exploration as interruption of the repeat motif has also been reported for other repeat expansion diseases like Friedreich's Ataxia or Huntington disease”

Line 330: ‘mobile CpG-island’ - this is good comparison, may use this wording earlier in the text, in Introduction section.
Response: We appreciate the suggestion and added the phrase to the Introduction section.
Introduction: “The SVA itself is highly methylated due to the high "GC" content (~60%) within the variable number tandem repeat (VNTR) region, also known as “mobile CpG-island”.”

Lines 345-346: (N-TAF1) - this abbreviation used only once, may remove it for better readability of the text.
Response: We have removed the abbreviation.

Line 362: ‘SVAf ‘ - what is lower case letter ‘f’, a typo?
Response: There are six different subfamilies within the SVA elements (SVAa SVAf). SVAe and SVAf are only present in the human genome. We have now specified this in the text “subfamily SVAf ”.

Line 379: ‘As a pilot, we..’ - use wording ‘pilot study’, not just ‘pilot’.
Line 388: ‘., title, Table’ - some redundant dots and word ‘title’.
Avoid ‘.,’ in the text, use ‘;’ sign instead.
Response: We thank the reviewer for both above suggestions and have changed the text accordingly.

This manuscript is a resubmission of an earlier submission. The following is a list of the peer review reports and author responses from that submission.

Round 1

Reviewer 1 Report

X-linked dystonia-parkinsonism is a Mendelian genetic disorder caused by SVA retrotransposon insertion in the TAF1 gene, which encodes a zinc-finger transcription factor TFIID. The number of hexanucleotide repeats (CCCTCT)n is inversely correlated to the age of onset in the disease. SVA insertion in TAF1 is associated with decreased TAF1 expression, and DNA methylation was suggested as a plausible or possible mechanism for the decreased expression.

  1. Confirmatory nature: The results in the Hexanucleotide Repeat Length are mainly confirmatory in nature and thus tarnish its originality.
  2. Non-conclusive results due to lack of power. For the methylation results, due to the small sample size (1 each for patient and for control), the confidence to make conclusions is reduced. Furthermore, the negative results should be discussed in relation to other positive correlational results such as Christin Krause et al, DNA Methylation as a Potential Molecular Mechanism in X-linked Dystonia-Parkinsonism , Movement Disorders, Vol. 35, No. 12, 2020.
  3. Lack of statistical analysis. Figure 3 should be analyzed with computational/bioinformatics/ softwares. One example is from Adam D. Ewing et al, Nanopore Sequencing Enables Comprehensive Transposable Element Epigenomic Profiling, Molecular Cell, 2020.
  4. Inadequate description of the statistical methods used: A whole section of “Statistical Methods” should be added and all the details added.
  5. Nanapore sequencing method(s): More details should be given. See (Authors’ own published works) Reyes et al, Brain Regional Differences in Hexanucleotide Repeat Length in X-Linked Dystonia-Parkinsonism, Neurology Genetics, 2021.
  6. Figure 3C is mislabeled. The control should be from Blood as well, not CRB (cerebellum).
  7. Line 231: Methylation ….. “were” analyzed. (Should use “was”.)
  8. Line 262: what is “the intern CpG island”?
  9. Line 325: Remove “Please add:”.
  10. Line 340-342: Delete “In this section, you can acknowledge any support given which is not covered 340 by the author contribution or funding sections. This may include administrative and technical sup-341 port, or donations in kind (e.g., materials used for experiments).”

Reviewer 2 Report

X-linked dystonia-parkinsonism (XDP) is an adult-onset neurodegenerative disorder characterized by progressive dystonia and parkinsonism.  A polymorphic (CCCTCT)n domain ranged from 30 to 55 happened in TAF1 as a genetic modifier of disease onset and expressivity were reported. However, due the limitations of NGS method, it is hard to characterize the TAF1 SVA insertion in XDP patients. In this manuscript, the authors applied an advanced technique for long reads sequencing called Nanopore to determine hexanucleotide repeat number, and CpG methylation of Cas9-targeted sequencing.  The manuscript is very well written and organized, which set a good example to study the disease related variations using Nanopore. There are some issues to be resolved before publication.

  1. The authors did not submit the nanopore data due to the patient’s confidentiality, while the author should provide the detailed information about the Nanopore sequencing, such as quality of reads, sequencing size and so on.
  2. In the results part, the author found “The overall MF across the SVA flanking region in the control sample 196 was at 0.83±0.17, which was lower than the patient-derived sample (exploratory Mann 197 Whitney U-test p<1×10-15)”. In addition, “The most pronounced difference was detected in enhancer sites six (mean MF patient: 207 0.91, mean MF control: 0.71) and nine (mean MF patient: 0.98, mean MF control: 0.70)”. But it is confusing that the author concluded in the abstract “There was no difference in genetic variability other than variations of the repeat motif between patients. 39 CpG methylation frequency (MF) in the SVA was high (mean MF=0.94, SD=±0.12). In silico predicted 40 enhancers flanking the SVA had a similar MF in a patient compared to a control.”
  3. The author found Nanopore sequencing found a mean of 42.77 (SD=±4.05) repeats, ranging from 33 to 54 in patients, why not compare this repeat number to the normal samples?
  4. In Figure 3 and 4, the significance of statistic test between normal and patients need provided if available.

Reviewer 3 Report

In their manuscript entitled “Elucidating hexanucleotide repeat number and methylation within the X-linked dystonia-parkinsonism (XDP)-related SVA retrotransposon in TAF1 with Nanopore sequencing”, the authors used Nanopore sequencing to test the effect of SVA retrotransposon insertion on the CpG methylation status of TAF1 loci and whether this could explain the decrease in TAF1 expression. Nanopore sequencing is used both to assess the (CCCTCT)n repeat length in PCR amplicons from 96 patients and to establish the CpG methylation status of TAF1 locus in one patient and one control blood sample through Cas9-targeted sequencing. Unfortunately, the CpG methylation was only assessed in one patient which is insufficient to draw any conclusion.

1/ Nanopore sequencing is being used by several groups to assess hexanucleotide repeat expansions. Indeed, several publications have already validated this technique, including one by Dr Trinh’s group where they used this technique on 2 XDP patients. In the present study, the technique is performed on 96 patients but the authors did not perform any correlation between the repeat length and the age of onset or the severity of the disease. Previous publications mentioned an inverse correlation between the number of repeats and the age of onset. This analysis should be performed and the results discussed.

2/ The mean coverage is 17645X with a SD of 12392x which shows a huge heterogeneity of the results. It would be more informative to present the distribution of the coverage and to select only the full size reads for the rest of the analysis. Also, it is not clear why the authors used both the super accuracy model and the fast model for the detection of the repeat length. It is well known that the fast model is less accurate than the Super accuracy one but the results obtained by the Fast Model seem to correlate better with the fragment analysis. The authors should elaborate on this point.  

3/ Fig 2 presents the number of deletions, insertions or substitutions in the repeat motif but the size of these indels is not mentioned nor the relevance of these interruptions in terms of disease severity. Also, considering that the number of repeats is different depending on the model used for the analysis, these results should be validated to exclude any artefact caused by the technique itself Indeed, Nanopore sequencing is known to accumulate errors such as indels.

4/ The CpG methylation status of one patient (blood and brain derived sample) and one control (blood derived) was determined using Cas9 targeted sequencing. The authors should explain better how the experiment was performed and especially why they use 4 crRNA to prepare each library.

Fig S1 shows a very uneven coverage, which explains the high Standard Deviation (858.2X) for a mean coverage of 397.5X over the whole region. It would be more informative to give the mean coverage for each library independently and especially for the library covering the 22kb region including the 12 enhancers.  

It would have been useful to have access to the raw data to be able to evaluate the quality of those data.

5/ CpG methylation is only assessed in one patient and one control. Therefore, these results can only be considered as preliminary and the authors should test at least 3 patients to consider drawing any conclusion.

Minor comments :

Abstract :

P1 L3 : Replace “an” by “a”

Introduction

P2, L54 : Replace “SVA TAF1” by “TAF1 SVA”

Material and Methods :

The authors should indicate the polymerase used to perform the long-range PCR as some DNA polymerase are known to accumulate errors in highly repetitive regions?